# Vocal signatures affected by population identity and environmental sound levels

**Amber D. Fandel** [ID]*, **Kirsten Silva, Helen Bailey**

Chesapeake Biological Laboratory, University of Maryland Center for Environmental Science, Solomons, MD, United States of America

* adfandel@gmail.com

## Abstract

Passive acoustic monitoring has improved our understanding of vocalizing organisms in remote habitats and during all weather conditions. Many vocally active species are highly mobile, and their populations overlap. However, distinct vocalizations allow the tracking and discrimination of individuals or populations. Using signature whistles, the individually distinct calls of bottlenose dolphins, we calculated a minimum abundance of individuals, characterized and compared signature whistles from five locations, and determined reoccurrences of individuals throughout the Mid-Atlantic Bight and Chesapeake Bay, USA. We identified 1,888 signature whistles in which the duration, number of extrema, start, end, and minimum frequencies of signature whistles varied significantly by site. All characteristics of signature whistles were deemed important for determining from which site the whistle originated and due to the distinct signature whistle characteristics and lack of spatial mixing of the dolphins detected at the Offshore site, we suspect that these dolphins are of a different population than those at the Coastal and Bay sites. Signature whistles were also found to be shorter when sound levels were higher. Using only the passively recorded vocalizations of this marine top predator, we obtained information about its population and how it is affected by ambient sound levels, which will increase as offshore wind energy is developed. In this rapidly developing area, these calls offer critical management insights for this protected species.

## Introduction

Since time immemorial, humans have observed oceans, lakes, and rivers, and the organisms in those environments have used sound to communicate. Human's ability to record sound under water, however, only date back to 1912 when an echo-ranging patent application was filed. Since then, the advancement of technology has allowed us to record, store, and process acoustic recordings both in real time and in post-processing. These advancements have also allowed for acoustic analyses in remote habitats and adverse weather conditions. Passive acoustic monitoring is especially valuable for monitoring vocalizing species and uses microphones (referred to as hydrophones in the underwater environment) to record vocalizing organisms. In the marine environment, one of the most vocal groups of organisms are marine mammals, which

**Data Availability Statement:** All relevant data are available on Dryad (https://doi.org/10.5061/dryad.9s4mw6mq0).

**Funding:** The Maryland Department of Natural Resources secured funding for this project from the Maryland Energy Administration's Offshore Wind Development Fund (Contract number 14-18-2420 MEA: HB, ADF, KS: energy.maryland.gov/Pages/Info/renewable/offshorewind.aspx). The views and conclusions contained in this document are those of the authors and should not be interpreted as representing the opinions or policies of the Maryland Department of Natural Resources, or the Maryland Energy Administration. Mention of trade names or commercial products does not constitute their endorsement by the state. The funders had no role in study design, data collection and analysis, decision to publish, or preparation of the manuscript.

**Competing interests:** The authors have declared that no competing interests exist.

produce a variety of calls. In addition to being able to detect their presence from these calls, some species have pod-specific calls [1–3], allowing the detection of specific population groups. In killer whales (*Orcinus orca*), family pods are differentiated based on the characteristics of their vocalizations [2].

The highly vocal and social bottlenose dolphin (*Tursiops runcates*) produces individually distinct calls called signature whistles [4, 5]. These signature whistles comprise more than half of the whistles that wild bottlenose dolphins produce [6, 7]. They are formed in the first year of the dolphin's life [8] and remain relatively stable over their lifetime [4, 8]. These unique whistles have been used to both identify [9, 10] and to estimate the number of individuals within small (< 100 individuals), resident [11–13] populations but less is known about their utility for larger, wide-ranging populations [2]. Distinct populations of bottlenose dolphins are known to vary the characteristics of their whistles based on sound levels [14, 15], but such variations in signature whistle characteristics have not yet been investigated.

The Atlantic coast of the USA is home to 19 different populations of bottlenose dolphins [16]. The region along the coasts of Delaware, Maryland, and Virginia, within the southern Mid-Atlantic Bight, is one of the least studied areas for bottlenose dolphins on the U.S. Atlantic coast. Previous studies in this region investigated the presence and temporal patterns of bottlenose dolphins [17–20] and documented their signature whistles [21]. This study aims to build upon that work and determine 1) a minimum estimate of the number of bottlenose dolphins in the Chesapeake Bay and waters of coastal Maryland, 2) whether there are site- or regionally-specific differences in signature whistle characteristics and how they relate to existing population boundaries, and 3) whether individual dolphins alter their signature whistles in response to ambient sound conditions. The findings from this study will help to determine bottlenose dolphin identities and whether these can be used to distinguish bottlenose dolphin populations using signature whistles. This work has implications for dolphin population management in the urbanized waterway of the Chesapeake Bay and in the Mid-Atlantic Bight where offshore wind energy development has been proposed.

## Methods

### Signature whistles as a measure of minimum abundance

Passive acoustic monitoring occurred at five sites: two in the southern coastal Mid-Atlantic Bight (blue triangles; Sites 1, 2, Coastal region), two in the Chesapeake Bay (black circles; Site 3, 4, Bay region), and one offshore in the southern Mid-Atlantic Bight (red square; Site 5, Offshore region; Fig 1). No permits were required for this work as there was no handling or interaction with the study species, but permission was obtained from the leaseholders of the wind energy area for the deployment of bottom-anchored autonomous recorders. Sites 1, 2, and 5 were located 12, 31, and 64 km east of Ocean City, Maryland, USA, respectively (Fig 1). Water depths at these sites ranged from approximately 20–42 m, and the acoustic recording instruments were deployed approximately 1 m above the ocean floor using bottom-anchored moorings [21, 22].

Recordings using an SM3M (Wildlife Acoustics, MA, USA) at Site 1 occurred between June and August in 2017 and July to October in 2018 (Table 1). At Site 2, recordings on an SM3M occurred between July and September of 2016, January to April and June to October in 2017, and June to December in 2018 (Table 1). Site 3 recordings occurred between June and August in 2018using the Snap acoustic recorder (Loggerhead Instruments, FL, USA). The Snap was also deployed at Sites 4 and 5 (Table 1). At Site 4 in the Potomac River, the Snap was deployed in approximately 3 m of water from June to September 2019 (Table 1). At Site 5, 64 km from the Maryland coastline, the Snap was deployed from July to September 2018 in 42 m of water

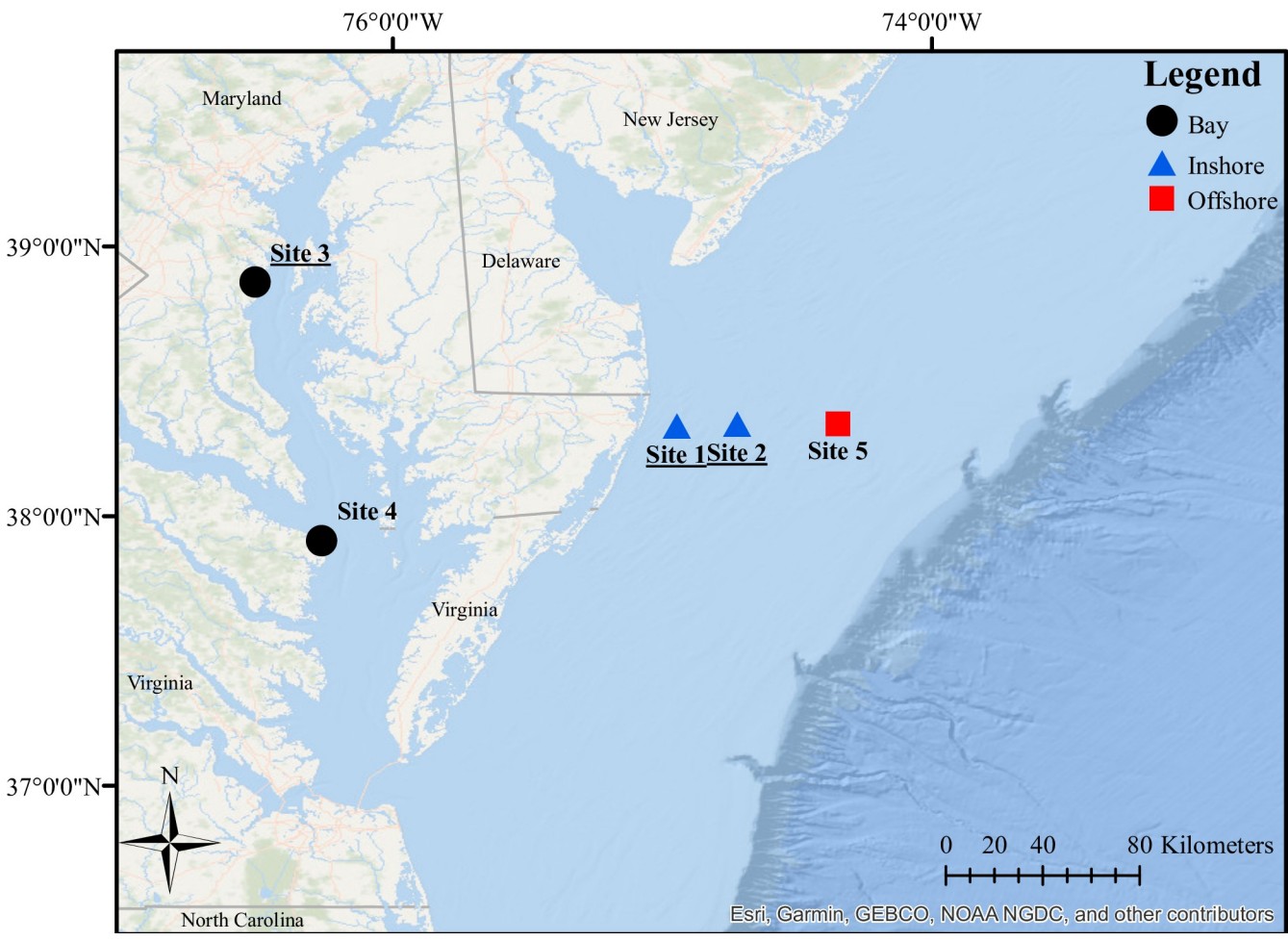

**Fig 1. Map of acoustic recording sites in the southern coastal Mid-Atlantic Bight, USA (blue triangles; Sites 1, 2), Chesapeake Bay (black circles; Site 3, 4), and offshore in the Mid-Atlantic Bight (red square; Site 5; following site nomenclature in Bailey et al. 2021 with additional sites).** Sites at which data were gathered in previous studies are underlined. Basemap images are the intellectual property of ESRI and are reprinted from Esri under a CC BY license with permission from Esri and its licensors, all rights reserved. Credits: ESRI, HERE, Garmin, GEBCO, OpenStreetMap, the GIS user community, NOAA NGDC, and other contributors.

(Table 1). At both sites, the Snap was approximately 1 m above the seabed sampling at 48 kHz with a sensitivity of 180.2 dB re 1V/μPa and a gain of 2.05 dB re 1μPa.

To extend battery life during the several months-long deployments, the Snap was duty-cycled for two minutes on and eight minutes off at Site 4 and one minute on and nine minutes

**Table 1. For each site and region, the recording period, total number of signature whistles detected, and total number and percentage of signature whistles selected for acoustic characteristic measurement.**

| Site | Region | Distance from shore | Recording period | Recording instrument and company |
|------|--------|---------------------|------------------|----------------------------------|
| 1 | Coastal | 12 km | June-Aug 2017 July-Oct 2018 | SM3M, Wildlife Acoustics |
| 2 | Coastal | 31 km | July-Sept 2016 Jan-April, June-Oct 2017 June-Dec 2018 | SM3M, Wildlife Acoustics |
| 3 | Bay | | May-Sept 2018 | Snap, Loggerhead Instruments |
| 4 | Bay | | June-Sep 2019 | Snap, Loggerhead Instruments |
| 5 | Offshore | 64 km | July- Sep 2018 | Snap, Loggerhead Instruments |

off at Site 5. At Site 4, a subsample of days were analyzed because of frequent dolphin presence. All recordings from Site 3 were manually analyzed due to very high sounds levels. At Sites 1, 2, and 5, the PAMGUARD Whistle and Moan Detector [22] was utilized to determine hours with possible dolphin presence. These hours were then manually searched for signature whistles with high signal-to-noise ratios. Signature whistles were manually identified using the SIGID criteria [10]-in which the same whistle repeated in a pattern of two or more whistles [6] within 1–10 s of one another and with a minimum length of 0.2 s [23, 24]. Whistle contours (shape of the whistle) were obtained using Beluga software (https://synergy.st-andrews.ac.uk/soundanalysis) within MATLAB (Math-Works, Natick, Massachusetts, USA). Whistles with low signal-to-noise ratio or abundant non-linear features [25] that obscured the shape of the whistle could not be included in the analysis.

Beluga contours were analyzed within the adaptive resonance theory neural network (ART-warp; vigilance threshold of 96%; [26]) to identify when a whistle reoccurred. Identical signature whistles were not considered reoccurrences if they occurred on the same day at the same site. A human analyst verified all whistle reoccurrences [21]. When the same signature whistle was identified at a different location or on a different day than the first instance, it was considered a reoccurrence of that individual signature whistle indicating the same dolphin had been detected again at a different location or time.

To confirm that the whistles detected were bottlenose dolphin, not common dolphin (*Delphinus delphis*), we tested the utility of the Real-time Odontocete Call Classification Algorithm (ROCCA) in PAMGUARD [27]. While other delphinid species are present in this region during the winter months, at the sites and times chosen, bottlenose dolphins were the most likely species present based on a previous analysis of visual sightings data in the study area [20].

However, in a total of 21 hours of recordings on 16 days during the summer at Site 1, ROCCA classified 61% of whistles as "Ambiguous" (n = 576), 13% as striped dolphin (*Stenella coeruleoalba*; n = 121*)*, 11% as bottlenose dolphins (n = 103), 8% as common dolphins (*Delphinus delphis*, n = 74), 4% as Clymene dolphin (*Stenella clymene*, n = 39*)*, 2 as Atlantic spotted dolphin (*Stenella frontalis*, n = 19), and 1% as Pantropical spotted dolphin (*Stenella attenuata*, n = 7). Clymene dolphins are typically found in waters deeper than 800 feet, and Pantropical spotted dolphins are not located in the Atlantic Ocean. Given the unreliability of these results, we were unable to confirm species identity using the acoustic data with ROCCA. We confirmed that similarly poor results were obtained in other studies using comparing ROCCA results to concurrent visual and passive acoustic observations in the Northwest Atlantic (in an email from S. Van Parijs, PhD (sofie.vanparijs@noaa.gov) in January 2021). Instead of relying on these automated detection methods, we used the spatial and temporal patterns determined in previous studies [20, 28] to determine which species were present and avoided sites and seasons in which there was likely to be overlap in species' presence.

## Acoustic characteristics of signature whistles

To determine the acoustic characteristics of signature whistles from each site, we selected a subset of 100 signature whistles from each site (or the number of unique signature whistles identified if <100). For each whistle, manual measurements were taken in Raven Pro 2.0 Interactive Sound Analysis Software (Cornell Lab of Ornithology, Center for Conservation Bioacoustics, Ithaca, New York, USA) of the duration, start, end, maximum, minimum, and delta frequencies (maximum minus minimum frequency), and number of local extrema (e.g. local minima and maxima) including the start and end of the whistle (similar to [12, 29]; Fig 2). The distribution of the signature whistles' characteristics was analyzed for normality using

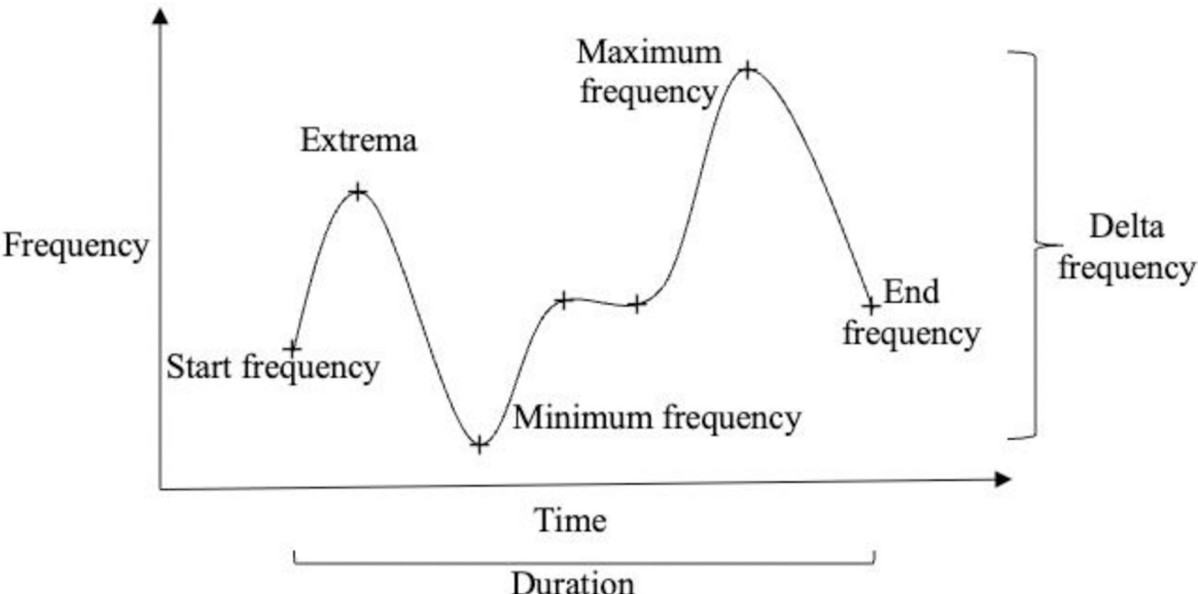

**Fig 2. Diagram of whistle showing characteristics measured including start, end, minimum, maximum, and delta frequencies, number of extrema, and duration.**

Shapiro-Wilk tests, and Box-Cox analyses (MASS package in R; [30]) were used to determine whether transformations of the characteristics were necessary for parametric analyses.

A multivariate analysis of variance (MANOVA) in R was utilized to determine whether the whistle characteristics (response variables) were related to the site (Sites 1–5) or region (Bay, Coastal, Offshore) at which the signature whistle was detected (explanatory variables). Post-hoc individual pairwise t-tests with Bonferroni adjusted p-values were used to investigate which and how signature whistle characteristics varied between sites and regions. The Boruta package in R [31] randomizes the data for each factor (called shadow data) then utilizes random forest models with the data and shadow data to assess factor importance in determining the identity (e.g. site or region) of the signature whistle. This model was employed to determine the relative importance of each signature whistle characteristic in that process. The maximum number of runs for the Boruta model was 10,000.

### Signature whistle variation in relation to ambient sound levels

To understand whether bottlenose dolphins alter their signature whistles in response to ambient sound levels, a subset of 100 signature whistles that reoccurred at least once, were selected for measurement (duration, start, end, maximum, minimum, and delta frequencies, number of local extrema). Ambient sound levels were calculated in MATLAB (MathWorks, Natick, Massachusetts, USA) as the relative broadband (up to 24 kHz, given the sampling rate of 48 kHz) root mean square sound pressure level (SPL; dB re 1µPa root-mean-square (rms)) during the recording in which the signature whistle occurred (two or five minutes in duration). Sound levels were also calculated for the duration of each acoustic deployment to determine overall ambient sound levels for each site. Generalized estimating equations-generalized linear models (GEEs) in R (geepack package in R; [32]) were used to determine whether and which changes in signature whistles characteristics were correlated with changes in ambient sound levels. The characteristics of 100 randomly selected signature whistles that re-occurred were modeled with the GEEs against the sound levels in which they occurred. The identity of the

**Table 2. For each site and region, the recording period, total number of signature whistles detected, and total number and percentage of signature whistles selected for acoustic characteristic measurement.**

| Site | Region | Recording period | Hours analyzed | Total number of unique SW | Number of SWs with measured characteristics | Percentage of total SW with measured characteristics (%) |
|------|--------|------------------|----------------|---------------------------|---------------------------------------------|----------------------------------------------------------|
| 1 | Coastal | June-Aug 2017, July-Oct 2018 | 268 | 1172 | 100 | 9 |
| 2 | Coastal | July-Sept 2016, Jan-April, June-Oct 2017, June-Dec 2018 | 521 | 327 | 100 | 31 |
| 3 | Bay | May-Sept 2018 | 2488 | 19 | 19 | 100 |
| 4 | Bay | June-Sep 2019 | 311 | 333 | 100 | 33 |
| 5 | Offshore | July- Sep 2018 | 415 | 37 | 37 | 100 |

signature whistle was treated as a cluster and an exchangeable correlation structure was used in the GEEs.

## Results

### Signature whistles as an indicator of minimum abundance

A total of 1,172 unique signature whistles were detected at coastal Site 1, 327 at coastal Site 2, and 19 at Site 3 in the Bay [21]. A total of 333 unique signature whistles were detected at Site 4 in the Potomac River in the Chesapeake Bay, and 37 at the offshore site, Site 5 (Table 2). An individuals dolphin's presence is indicated by the presence of its signature whistle, so these counts of unique signature whistles represent a minimum abundance of bottlenose dolphins occurring at each site.

### Acoustic characteristics of signature whistles

For the subset of signature whistles for which whistle characteristics were measured Table 1), the distribution of these characteristics (start, end, maximum, minimum, and delta frequencies, duration, number of local extrema) were non-parametric (Shapiro-Wilk test; $p < 0.05$). Following Box-Cox analyses and appropriate transformations, the data were tested again for normality, but only the start, minimum, and delta frequencies were normally distributed. Because the distribution of end frequencies remained non-parametric after transformation, a square root (sqrt) transformation was utilized for consistency with other frequency measures.

A MANOVA indicated that start ($F_{4,352} = 2.71$, $p = 0.03$), end ($F_{4,352} = 4.52$, $p = 0.01$), and minimum frequencies ($F_{4,352} = 10.3$, $p < 0.01$) as well as duration ($F_{4,352} = 9.36$, $p < 0.01$) and number of extrema ($F_{4,352} = 7.28$, $p < 0.01$), or complexity, of signature whistles varied significantly by site. Pairwise t-tests indicated that the start and end frequencies of signature whistles were higher (Fig 3A and 3B) and delta frequencies were lower (Fig 3D) at Site 5 compared to Site 4. Minimum frequencies of signature whistles at Site 5 were higher than all other sites (Fig 3C), and signature whistles from Site 2 had significantly longer durations than whistles from Site 1, 3, and 4 (Fig 3E). The number of extrema were significantly fewer in signature whistles from Site 5 compared to any site other than Site 3, and there were significantly more local extrema at Site 1 compared to Site 2 (Fig 3F).

A MANOVA also indicated that all characteristics of the signature whistles except the delta frequencies varied significantly by region. Offshore signature whistles had significantly higher start, end, and minimum frequencies compared to those from the Coastal and Bay regions (Fig 4A–4C). The duration of signature whistles from the Bay were shorter than those from the Coastal or Offshore regions (Fig D, and Offshore signature whistles had significantly fewer extrema than the Bay or Coastal whistles (Fig 4E).

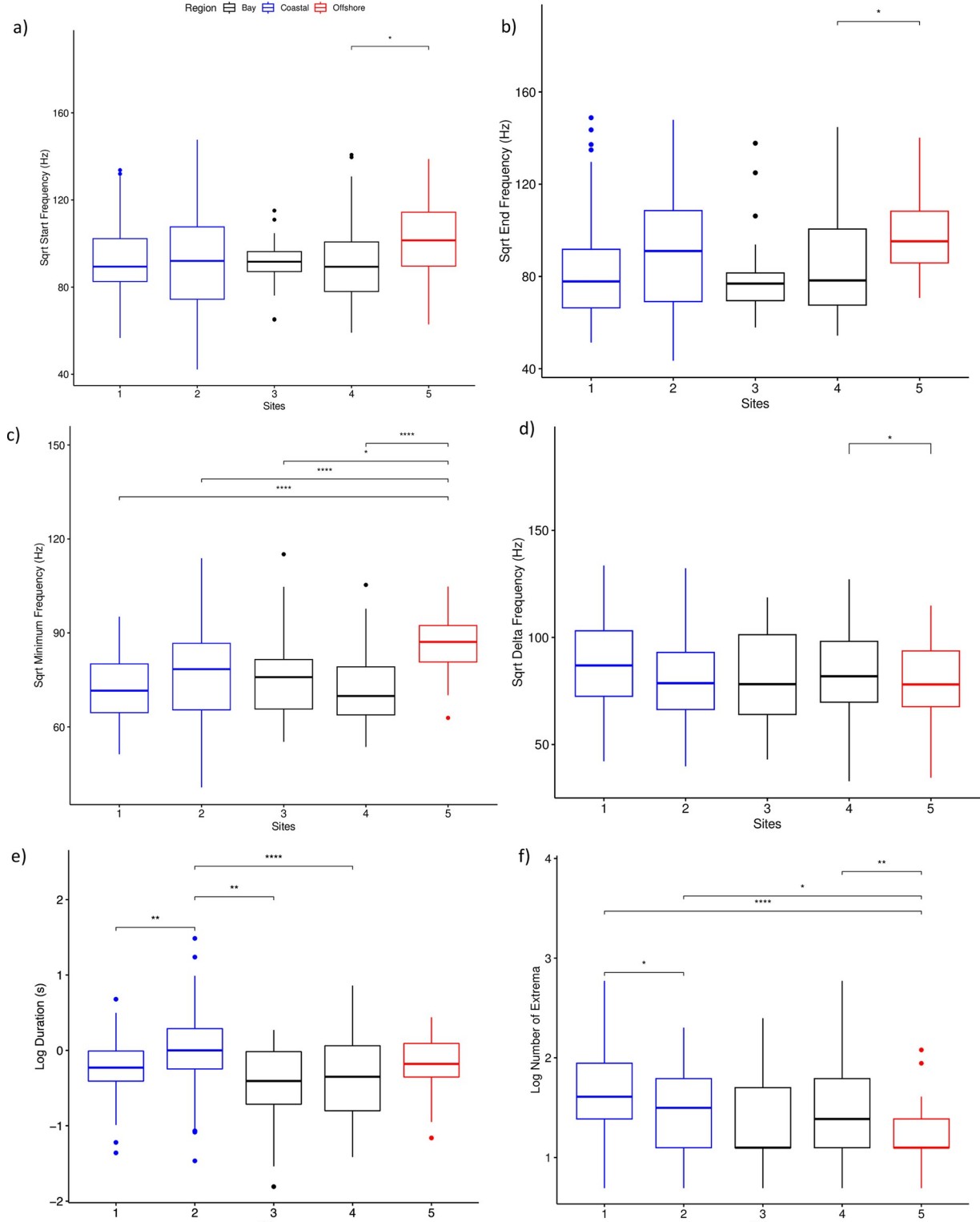

**Fig 3.** Standard boxplot of the a) square root (sqrt) of the start frequency, b) sqrt of the end frequency, c) sqrt of the minimum frequency, d) sqrt of the delta frequency, e) log of the duration, and f) log of the number of extrema for signature whistles at each site. Asterisks indicate an adjusted p-value (Bonferonni method) of less than 0.05 in pairwise t-tests. Blue boxes indicate sites in the Coastal region, black are sites in the Chesapeake Bay, and red is the site in the Offshore region.

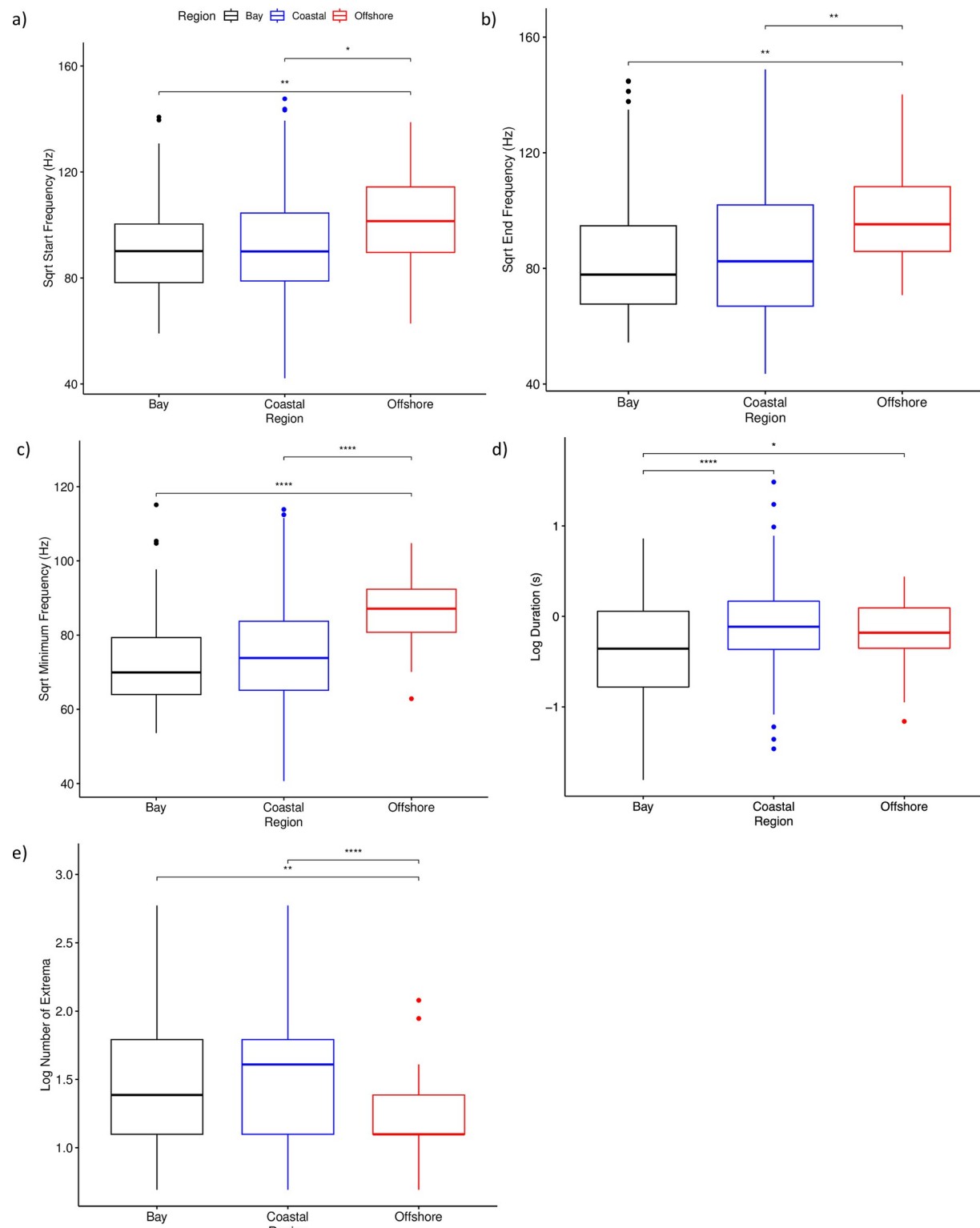

**Fig 4.** Standard boxplot of the a) square root (sqrt) of the start frequency, b) sqrt of the end frequency c) sqrt of the minimum frequency, d) log of the duration, and e) log of the number of extrema for signature whistles in each region. Asterisks indicate an adjusted p-value of less than 0.05 (Bonferonni method) in pairwise t-tests.

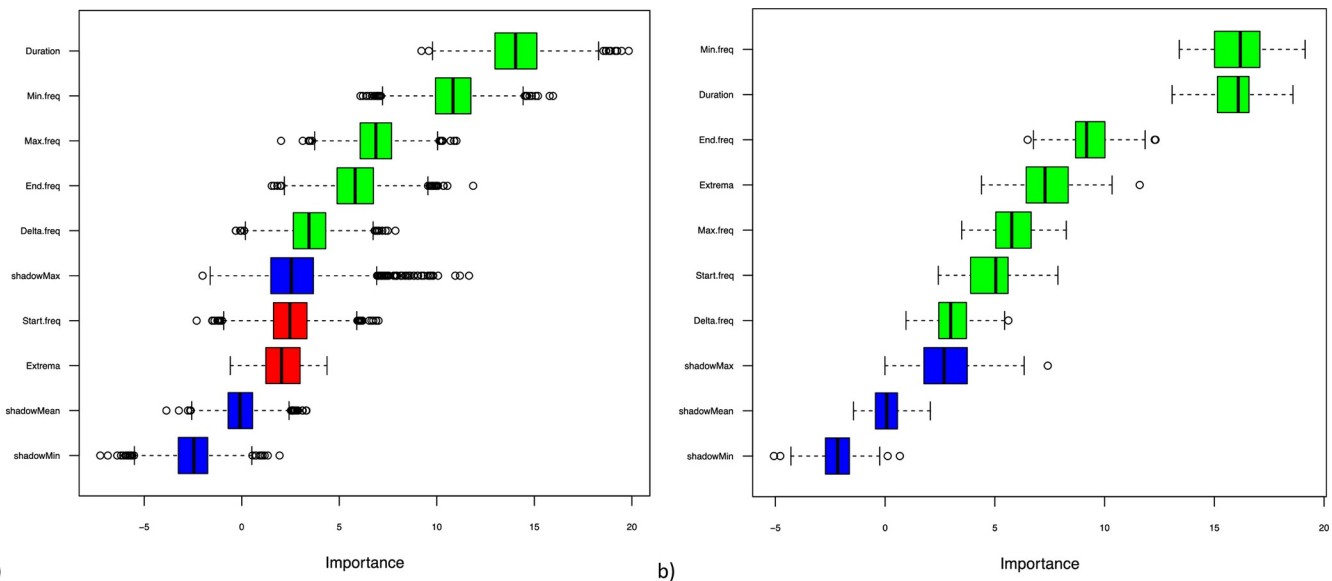

**Fig 5.** Results of the Boruta random forest model indicating the ranked importance of signature whistle characteristics in determining a whistle's (a) site and (b) regional identities. Green boxes indicate important characteristics, red indicates characteristics that decreased the performance of the model, and the blue are shadow factors used to test the model. Shadow factors are shuffled duplicate copies of factors added to remove correlations between variables.

All signature whistle characteristics were deemed important for determining site identity, but minimum frequency and duration were the most important factors (Fig 5A). The duration, minimum, maximum, end, and delta frequencies were the important characteristics in determining the regional identity of signature whistles (Fig 5B). When the start frequency and number of extrema were included in the model, the model was less accurate when determining the region to which a signature whistle belonged (Fig 5B).

Median daily sound levels in all sites and regions were significantly different from one another (p < 0.01). Site 1 was loudest followed by Site 5, Site 2, Site 3, and Site 4 (Fig 6). The Offshore region had the second highest daily median ambient sound levels (Fig 6) and the highest median daily minimum sound levels (127 dB re 1μPa, sd = 0.2 dB). Site 4 had the largest range in sound levels (Fig 6).

## Signature whistle variation in relation to ambient sound levels

In total, 201 unique signature whistles reoccurred a total of 252 times at the five different sites (Fig 7). Reoccurrences of whistles indicate that dolphins were detected at different sites on different days. Signature whistles most often reoccurred at the inshore sites, Sites 1 and 2 (n = 49; 15% of all unique signature whistles from Site 2) and between the most inshore site, Site 1, and the lower Bay site, Site 4 (n = 30, 9% of unique signature whistles from Site 4; Fig 3).

Of the 19 unique signature whistles detected at the upper Bay site (Site 3), 26% (n = 5) were also detected at the lower Bay site (Site 4; Fig 7). At the lower Bay site, 24 of the total 333 unique signature whistles identified reoccurred 27 times within the site on different days indicating that dolphins may be spending several days within the Bay and were detected as they entered and exited the Bay.

Of the 37 unique signature whistles detected at Site 5, only three signature whistles reoccurred within the site. Signature whistles from the offshore Site 5 were also detected three times at Site 2 (two of which were the same whistle; 5% of all unique signature whistles from Site 5) and one was detected at Site 1 (3%; Fig 7).

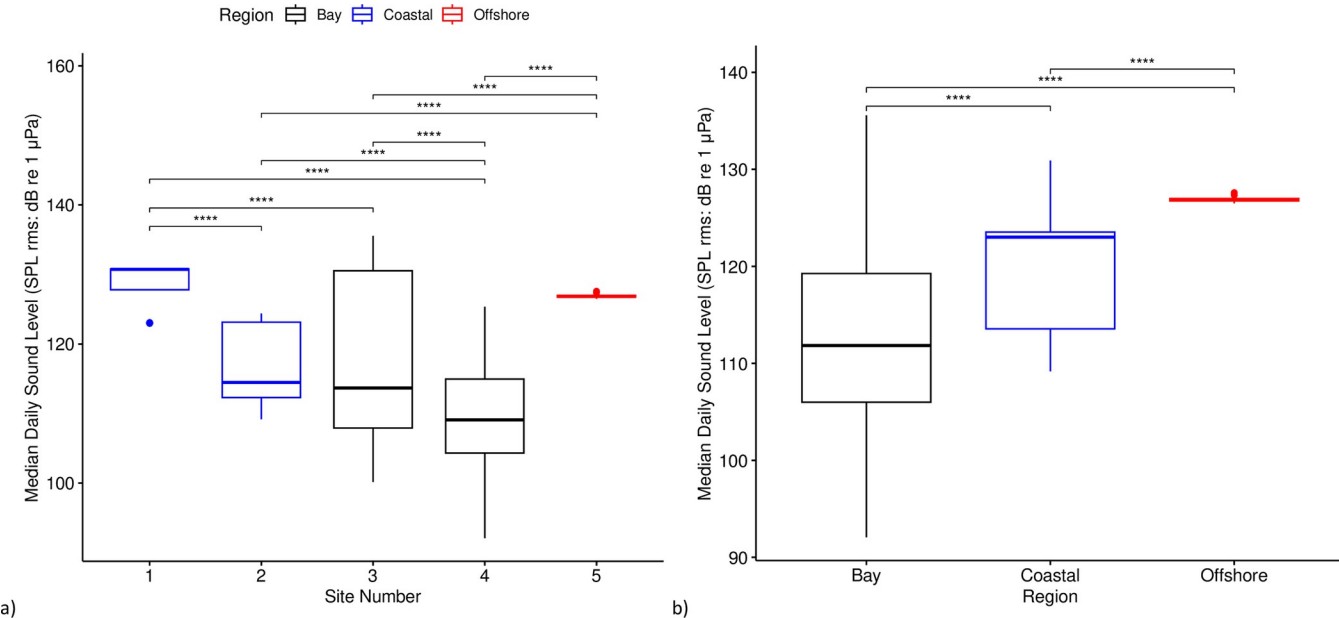

**Fig 6.** Median daily sound levels (SPL rms: dB re 1μPa) at each site (a) and region (b). Asterisks indicate an adjusted p-value (Bonferonni method) of less than 0.05 in the pairwise t-tests.

The random selection of 100 unique signature whistles that were analyzed to determine how signature whistle characteristics varied at differing sound levels included 230 total whistle occurrences: 132 whistles at Site 1, 49 at Site 2, 9 at Site 3, 36 at Site 4, and 4 at Site 5. When changes in signature whistle characteristics were modeled in relation to the ambient sound levels during which they occurred, only the duration of the signature whistles significantly varied in relation to changing ambient sound levels, decreasing as sound levels increased (Estimate = -0.005, S.E. = 0.003, Wald = 4.08, p = 0.04). All other signature whistle characteristics did not vary between reoccurrences when sound levels varied.

## Discussion

We determined that bottlenose dolphins reoccurred through time and space and that the characteristics of bottlenose dolphin signature whistles varied significantly by site and region. This may represent acoustic differences amongst the signature whistles of estuarine, coastal, and offshore bottlenose dolphin populations. Notably, the same individual shortened its signature whistle in relation to increased ambient sound level. This is the first quantitative investigation into the effects of location and ambient sound levels on individually identifiable dolphin calls.

Characterizing signature whistles may allow for the acoustic differentiation of populations or stocks. This ability is especially valuable in areas of population overlap such as the Mid-Atlantic Bight and the Chesapeake Bay. Our study area is within the home range of up to four different populations of bottlenose dolphins; the Western North Atlantic (WNA) Northern Migratory Coastal, WNA Southern Migratory Coastal, Northern North Carolina Estuarine System, and the WNA Offshore stocks. The numerous differences in the acoustic characteristics of signature whistles from bottlenose dolphins in the Offshore region (Site 5, 42 m depth) and lack of reoccurrences of these whistles in the Chesapeake Bay suggests that these dolphins may be from the WNA Offshore stock. The movement patterns and number of individuals in this offshore population are poorly understood. Signature whistle analysis may allow us to

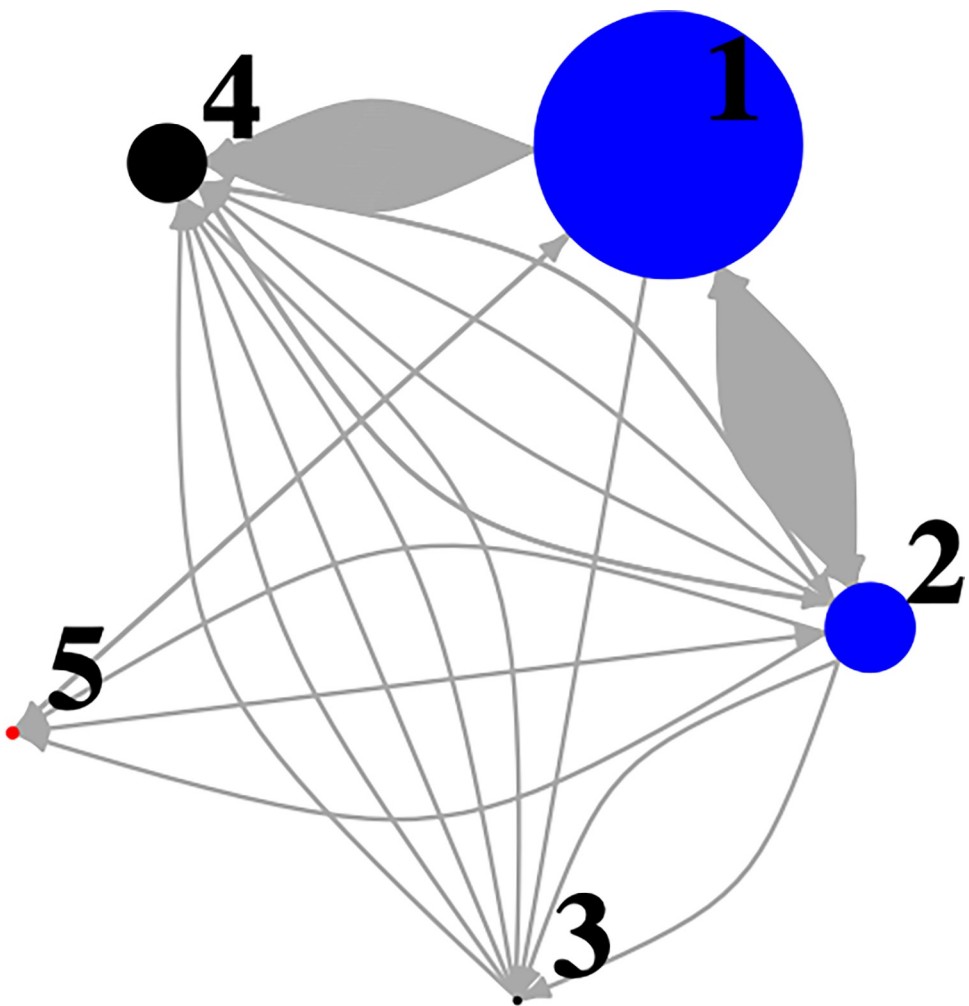

**Fig 7. Network diagram showing signature whistle reoccurrences (gray lines) between sites (circles labeled with site number) within the study area.** Blue nodes indicate sites in the Coastal region, black indicate sites in the Chesapeake Bay, and red indicates a site in the Offshore region. Size of the node indicates the number of signature whistles detected at each site.

track individuals over time and space, which is especially valuable in this highly urbanized area that is also slated for offshore renewable energy development.

Large numbers of bottlenose dolphins seasonally inhabit the Chesapeake Bay and Mid-Atlantic region [19, 33], and acoustic niche partitioning [34, 35] in signature whistles may be occurring [15, 36] due to population overlap. However, it is still unclear to which population (s) the bottlenose dolphins detected in the Bay belong. Population-level variation in vocal calls (i.e. vocal dialects) have been detected in multiple species [2, 37–39] and are likely utilized in this highly vocal and mobile species. Some signature whistles, but not all, detected in the Chesapeake Bay (Sites 3 and 4) also occurred in the Coastal region (Sites 1 and 2), suggesting that the Chesapeake Bay may be a mixing ground for multiple populations including the WNA Northern Migratory Coastal, WNA Southern Migratory Coastal, and Northern North Carolina Estuarine System stocks. It also indicates that the ranges of these populations may be more extensive than previously thought and re-consideration of stock delineations may be necessary.

The importance of identifying the stock identity of dolphins in the Chesapeake Bay is further necessitated by the evidence of calves [33]. To understand to which stock the calves may belong, the study area could be expanded to the north and to the south, extending into the known ranges of other populations of this species. This would offer the opportunity to record signature whistles from individuals that have been photo-identified. Additionally, the feasibility of using signature whistles and their characteristics for stock assignment could then be determined. This would allow a better understanding of stock structure, which is vital for management and consideration during environmental impact assessments.

The importance of signature whistles in communicating an individual's identity may also necessitate its acoustic differentiation from other whistle types. Signature whistles at Site 2 were generally longer (1.13 s), had lower minimum (5966 Hz), higher maximum (12750 Hz), and larger delta frequencies (6784 Hz) than a random selection of whistles from the same site (0.40 s, 6792 Hz, 100075 Hz, 3282 Hz; [29]). This dissimilarity of characteristics between a random selection of whistles and the signature whistles from Site 2 suggests that there may be some acoustic properties of signature whistles (in addition to their repeated pattern) that indicates their identity as signature whistles.

High ambient sound levels (> 120 dB re 1µPa, [40]) were recorded in this study area, and the duration of dolphin signature whistles decreased as ambient sound levels increased. Because our calculation of ambient sound levels did not exclude other whistles, higher sound levels could also be indicative of more whistles within the 2 or 5-minute recordings. Additionally, because sound with frequencies above 24 kHz could not be detected in our recordings, higher frequency signature whistles would have been missed in our analyses [41]. While an increase in maximum frequency in response to higher sound levels was found in some studies [29, 42, 43], the maximum frequency of some signature whistles in this region were high [8] and at the upper limit of our sampling frequency. Thus, alternative acoustic compensation strategies such as shortening the duration of signature whistles may have been necessary [44–48]. These compensation strategies, however, may not be adequate to avoid signal masking [49].

Offshore, where median daily ambient sound levels were highest, signature whistles were higher frequency (start, end, minimum) and simpler (with fewer extrema). Whistles with higher frequencies [29, 43] and fewer extrema [29, 42] are common where low frequency (generally below 3 kHz) ambient sound levels are higher. Large vessels and high sea states are the primary contributors to consistently high ambient sound levels at this location near the vessel separation schema leading into the busy Port of Wilmington, Delaware. Dolphins of this region are likely utilizing higher frequency whistles to avoid acoustic masking from vessel traffic [50, 51].

This study found that the characteristics of bottlenose dolphin signature whistles differed significantly between sites and regions. Understanding how signature whistles vary by location may delineate the distribution of specific populations in the region [52]. In addition, analyzing reoccurrences of bottlenose dolphins signature whistles indicated that they adjusted the duration of their signature whistles in relation to the ambient sound environment in which they were emitted, possibly to compensate for the increased noise and reduce the likelihood of interruption of their call. Increasing sound levels in the ocean may continue to cause changes in dolphin calls leading to information loss [53]. Signal masking could also reduce cohesion amongst dolphin groups, including mother-calf pairs, who use signature whistles to facilitate reunions [7].

Sound levels in this region will rise due to development of offshore wind energy, coastal development, bridge construction, and increasing vessel traffic. It is particularly important to understand how individuals from each population may be exposed to sound from various human activities and how they may be impacted by masking of calls and potential disruption

to their communication, all of which could affect their fitness. The use of signature whistles to track individuals can also aid in determining how long individuals take to return to an area after exposure to elevated sound levels. Such information will be vital for management of this region to determine the potential effects of, responses to, and recovery from anthropogenic sound pollution [52].

## Acknowledgments

Thank you to Elizabeth Gryzb, Elizabeth McDonald, and Aimee Hoover at the Chesapeake Biological Laboratory for their initial analyses, Matt Ogburn at the Smithsonian Environmental Research Center for assistance with obtaining recordings, and Aaron Rice at Cornell University.

## Author Contributions

**Conceptualization:** Amber D. Fandel, Helen Bailey.

**Data curation:** Amber D. Fandel, Kirsten Silva.

**Formal analysis:** Amber D. Fandel.

**Funding acquisition:** Helen Bailey.

**Investigation:** Amber D. Fandel, Kirsten Silva.

**Methodology:** Amber D. Fandel, Kirsten Silva, Helen Bailey.

**Project administration:** Helen Bailey.

**Supervision:** Helen Bailey.

**Visualization:** Amber D. Fandel.

**Writing – original draft:** Amber D. Fandel.

**Writing – review & editing:** Kirsten Silva, Helen Bailey.

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
