## [Decision Letter · Decision Letter 0]

11 May 2023

PONE-D-23-09444Vocal signatures affected by population identity and environmental sound levelsPLOS ONE

Dear Dr. Fandel,

Thank you for submitting your manuscript to PLOS ONE. After careful consideration, we feel that it has merit but does not fully meet PLOS ONE’s publication criteria as it currently stands. Therefore, we invite you to submit a revised version of the manuscript that addresses the points raised during the review process.

ACADEMIC EDITOR:Please carefully consider the comments and suggestions of changes made by both reviewers when preparing a revised version of your work. Especially, better describe the methods used at each study site and properly acknowledge the possible drawbacks of not using the same methodology at the discussion.==============================

We look forward to receiving your revised manuscript.

Kind regards,

Vitor Hugo Rodrigues Paiva, Ph.D.

Academic Editor

PLOS ONE

Journal Requirements:

"The Maryland Department of Natural Resources secured funding for this project from the Maryland Energy Administration’s Offshore Wind Development Fund (Contract number 14-18-2420 MEA: HB, ADF, KS: energy.maryland.gov/Pages/Info/renewable/offshorewind.aspx). The views and conclusions contained in this document are those of the authors and should not be interpreted as representing the opinions or policies of the Maryland Department of Natural Resources, or the Maryland Energy Administration. Mention of trade names or commercial products does not constitute their endorsement by the state."

"Thank you to Elizabeth Gryzb, Elizabeth McDonald, and Aimee Hoover at the Chesapeake Biological Laboratory for their initial analyses, Matt Ogburn at the Smithsonian Environmental Research Center for assistance with obtaining recordings, and Aaron Rice at Cornell University. The Maryland Department of Natural Resources secured funding for this project from the Maryland Energy Administration’s Offshore Wind Development Fund (Contract number 14-18- 2420 MEA). Amber Fandel received support from Maryland Sea Grant under award NA22OAR4170020 from the National Oceanic and Atmospheric Administration, U.S. Department of Commerce. The views and conclusions contained in this document are those of the authors and should not be interpreted as representing the opinions or policies of the Maryland Department of Natural Resources, or the Maryland Energy Administration. Mention of trade names or commercial products does not constitute their endorsement by the state."

"The Maryland Department of Natural Resources secured funding for this project from the Maryland Energy Administration’s Offshore Wind Development Fund (Contract number 14-18-2420 MEA: HB, ADF, KS: energy.maryland.gov/Pages/Info/renewable/offshorewind.aspx). The views and conclusions contained in this document are those of the authors and should not be interpreted as representing the opinions or policies of the Maryland Department of Natural Resources, or the Maryland Energy Administration. Mention of trade names or commercial products does not constitute their endorsement by the state."

6. We note that Figure 1 in your submission contain [map/satellite] images which may be copyrighted. All PLOS content is published under the Creative Commons Attribution License (CC BY 4.0), which means that the manuscript, images, and Supporting Information files will be freely available online, and any third party is permitted to access, download, copy, distribute, and use these materials in any way, even commercially, with proper attribution. For these reasons, we cannot publish previously copyrighted maps or satellite images created using proprietary data, such as Google software (Google Maps, Street View, and Earth). For more information, see our copyright guidelines: http://journals.plos.org/plosone/s/licenses-and-copyright.

Reviewers' comments:

Reviewer's Responses to Questions

**Comments to the Author**

1. Is the manuscript technically sound, and do the data support the conclusions?

Reviewer #1: Partly

Reviewer #2: No

2. Has the statistical analysis been performed appropriately and rigorously? 

Reviewer #1: Yes

Reviewer #2: Yes

3. Have the authors made all data underlying the findings in their manuscript fully available?

Reviewer #1: Yes

Reviewer #2: No

4. Is the manuscript presented in an intelligible fashion and written in standard English?

Reviewer #1: Yes

Reviewer #2: Yes

5. Review Comments to the Author

Reviewer #1: Comments on Reviewed Manuscript.

Title: Vocal signatures affected by population identity and environmental sound levels

ID No: PONE-D-23-09444

Summary: The manuscript proposes methods to estimate the minimum numbers of the bottlenose dolphins (bd) in the Chesapeake Bay and waters of coastal Maryland and determine differences in the vocalizations of the bd from these sites (if they are site-specific) by comparing them with the vocalizations of bd from other regions. Also, the authors checked if individual bd change their signature in response to ambient sound conditions. Existing models on R are used for the analysis of recorded acoustics data.

Comments:

Kindly find below my comments/suggestions:

1. Line 33-33. This looks like an incomplete statement. I would rather suggest you state that passive acoustic monitoring (PAM) is a valuable technique for monitoring vocalizing species. Also, a sentence or 2 to define PAM may be helpful. This open-access review article on marine mammal vocalization may be helpful https://ieeexplore.ieee.org/abstract/document/9110497

2. There are inconsistencies in the reference style adopted in the manuscript. Some seem APA (line 36 (Janik & Sayigh, 2013)), while others seem to be IEEE line 73 (14), Line 90 (16,17). Authors should stick to one (approved by the journal).

3. Introduction: a little more details should be given on PAM. The characteristics of dolphins' signatures and how they differ on an individual basis should be provided.

4. Figure 1: A clearer map of the study area should be provided. Also, in the explanation of the areas being investigated in this study, the authors should clearly state the two regions on the map. Also, did the author carry out new acoustic recordings for all regions or it uses already available data for some of the sites (as seems to be the explanation on Lines 72-94)? A distinction between sites/regions, as they relate to the manuscript, would make the manuscript clearer to readers.

5. What does 14 mean in line 73?

6. Line 86-87 states: Signature whistles were considered reoccurrences when they occurred on different days or in different locations.

This needs to be explained further.

7. Lines 121-130 state that: To understand whether bottlenose dolphins alter their signature whistles in response to……

How were the changes in ambient level determined/measured concerning the selected signature whistles? This needs to be buttressed further (particularly in the discussion of the results).

8. Results section: Lines 134-137 need further clarification, with further details on the results in Table 1.

9. Figures 2-6: Clearer figures (particularly the labels & values) should be provided. The present Figures are not readable!

10. Line 149: sqrt transformation, write in full at 1st mention.

11. Line 152 7 others that follow. What are the units of the frequencies and other parameters?

12. Lines 295-298: from ‘These findings…..to the end. The statement looks ambiguous and incomplete. Consider rephrasing with the clear meaning of the intended message. In motivating the need for effective management of the ocean due to increasing activities, more details should be provided by citing relevant related literature.

Reviewer #2: I appreciate the dataset that was curated for this manuscript and know all of the hard work that goes into these sorts of large data analyses. Unfortunately, I do not recommend this manuscript for publication at this time. The main reasons are with the methodology: the methods are not the same for all sites, the methods are not described in enough detail for SIGID, Artwarp thresholds, how many of the same whistle type were allowed at each site, the random forrest classifiers used to classify bottlenose from common dolphins whistles, and the low frequency sampling rate assessing things like maximum whistle frequency and broadband ambient noise levels. I think that with some care there can be some interesting findings from the robust dataset if the authors are able to consider these points.

Line Item Edits

Line 35: This is worded very strangely, consider reworking to just say that marine mammals are very vocal.

Line 38: Caldwell and Caldwell 1965 founded signature whistles not Sayigh 1990

Line 80. Why were all of these methods different? 8:2 and 9:1 duty cycle, visual observations compared to the whistle and moan detector?

Please include your own methods on identifying signature whistles from recordings using SIGID and Artwarp. These methods need to stand alone without needing to refer to other papers for operational definitions and thresholds used. For example what similarity threshold did you use to consider a signature whistle a reoccurrence.

A 48 kHz sampling rate gives a 24kHz Nyquist cut off. Dolphin whistles can range far above this range (up to over 40kHz). This needs to be clearly identified as a limitation of the study and may have resulted in distorted whistle feature data.

Line 84 you switch to using numbered citations instead of name and date?

Line 94: I am concerned about the species classification portion of this. You say you used ROCCA for species classification but it performed poorly (no data) and then say you used random forest models to confirm species identity but don’t provide any information on those models, their previous success rate or how you confirmed that those were successful here.

When you identified a signature whistle, did you analyze multiple iterations of that whistle or just 1 example? What if one dolphin is hanging around one site for a long time whistling, does their one unqiue signature whistle end up contributing much of the whistle data?

Line 125- What is a ‘clear’ signature whistle? Sportelli et al 2022 found that over 50% of signature whistles have non-linear features… Does this mean you did not include any whistles that had NLP?

Line 129- I am having a hard time with calling the ambient sound levels ‘broad band’ up to 24kHz when dolphin hearing goes into the 100s of kHz. Arguably there could have been any number of anthropogenic sound sources above the recording range.

Line 159- The word site keeps getting pasted into the sentence

Discussion- what was the point of identifying signature whistles if the question was based on whistle features changing based on population, location and or ambient noise? A discussion of why not include all whistles would be helpful.

If the signature whistle features were significantly different than a random selection of whistles at the same site, wouldn't this be an important result to suggest that the differences between sites may not be due to sites or ambient noise but just based on the fact that the different dolphins have different signature whistles?

Line 276: Your recording devices did not go up to the highest recorded whistle frequencies. See Jones et al., 2020 for a review or Kaplan and Reiss 2017 or Hiley et al. 2017 for evidence of whistles into the 40s of kHz.

6. PLOS authors have the option to publish the peer review history of their article (what does this mean?). If published, this will include your full peer review and any attached files.

Reviewer #1: No

Reviewer #2: No

---

## [Author Response · Author response to Decision Letter 0]

21 Aug 2023

Review Comments to the Author

Reviewer #1: Comments on Reviewed Manuscript.

Title: Vocal signatures affected by population identity and environmental sound levels

ID No: PONE-D-23-09444

Summary: The manuscript proposes methods to estimate the minimum numbers of the bottlenose dolphins (bd) in the Chesapeake Bay and waters of coastal Maryland and determine differences in the vocalizations of the bd from these sites (if they are site-specific) by comparing them with the vocalizations of bd from other regions. Also, the authors checked if individual bd change their signature in response to ambient sound conditions. Existing models on R are used for the analysis of recorded acoustics data.

Comments:

Kindly find below my comments/suggestions:

1. Line 33-33. This looks like an incomplete statement. I would rather suggest you state that passive acoustic monitoring (PAM) is a valuable technique for monitoring vocalizing species. Also, a sentence or 2 to define PAM may be helpful. This open-access review article on marine mammal vocalization may be helpful https://ieeexplore.ieee.org/abstract/document/9110497

• Thank you for the suggestion. A sentence on passive acoustic monitoring has been added.

“Passive acoustic monitoring uses microphones (referred to as hydrophones in the underwater environment) to detect the sounds of vocalizing organisms.” (line 35-36)

2. There are inconsistencies in the reference style adopted in the manuscript. Some seem APA (line 36 (Janik & Sayigh, 2013)), while others seem to be IEEE line 73 (14), Line 90 (16,17). Authors should stick to one (approved by the journal).

• Thank you for this correction. All citations are now Vancouver. 

3. Introduction: a little more details should be given on PAM. The characteristics of dolphins' signatures and how they differ on an individual basis should be provided.

• I have inserted “Passive acoustic monitoring uses microphones (referred to as hydrophones in the underwater environment) to detect the sounds of vocalizing organisms.” (line 35-36) and “These signature whistles comprise more than half of the whistles that wild bottlenose dolphins produce (7,8). They are formed in the first year of the dolphin’s life (9) and remain relatively stable (5,9).” (line 42-44)

4. Figure 1: A clearer map of the study area should be provided. Also, in the explanation of the areas being investigated in this study, the authors should clearly state the two regions on the map. Also, did the author carry out new acoustic recordings for all regions or it uses already available data for some of the sites (as seems to be the explanation on Lines 72-94)? A distinction between sites/regions, as they relate to the manuscript, would make the manuscript clearer to readers.

• The resolution of the map has been increased and I have also underlined the sites from previous studies. I have also included additional details about recordings from previous studies to supplement Table 1. “Previous recordings at Site 1 occurred between June and August in 2017 and July to October in 2018. At Site 2, recordings occurred between July and September of 2016, January to April and June to October in 2017, and June to December in 2018. Site 3 recordings occurred between June and August in 2018.” (line 82-85)

5. What does 14 mean in line 73?

• This was an incorrectly formatted reference. Thank you for the correction. 

6. Line 86-87 states: Signature whistles were considered reoccurrences when they occurred on different days or in different locations.

This needs to be explained further.

• I have explained this further so the line reads “When the same signature whistle was identified at a different location or on a different day than the first instance, it was considered a reoccurrence of that individual signature whistle indicating the same dolphin had been detected again.” (line 110-113)

7. Lines 121-130 state that: To understand whether bottlenose dolphins alter their signature whistles in response to……

How were the changes in ambient level determined/measured concerning the selected signature whistles? This needs to be buttressed further (particularly in the discussion of the results).

• To clarify this point, I have moved up the sentence at the end of this paragraph so it now reads “Ambient sound levels were calculated in MATLAB (MathWorks, Natick, Massachusetts, USA) as the relative broadband (up to 24 kHz, given the sampling rate of 48 kHz) sound pressure level (SPL; dB re 1µPa root-mean-square (rms)) during the recording in which the signature whistle occurred (two or five minutes in duration). Sound levels were also calculated for the duration of each acoustic deployment to determine overall ambient sound levels.” (line 159-164) and in the results “When changes in signature whistle characteristics were modeled in relation to the ambient sound levels during which they occurred, only the duration of the signature whistles significantly varied in relation to changing ambient sound levels, decreasing as sound levels increased (Estimate = -0.005, S.E. = 0.003, Wald = 4.08, p = 0.04). All other signature whistle characteristics did not vary between reoccurrences when sound levels varied. (line 263-268)

8. Results section: Lines 134-137 need further clarification, with further details on the results in Table 1.

• We have added context to these lines to read: “A total of 1,172 unique signature whistles were detected at coastal Site 1, 327 at coastal Site 2, and 19 at Site 3 in the Bay (18). A total of 333 unique signature whistles were detected at Site 4 in the Potomac River in the Chesapeake Bay, and 37 at the offshore site, Site 5 (Table 1). An individuals’ dolphin’s presence is indicated by the presence of its signature whistle, so these counts of unique signature whistles represent a minimum abundance of bottlenose dolphins occurring at each site.” (lines 174-179) and “For the subset of signature whistles for which whistle characteristics were measured Table 1), the distribution of these characteristics (start, end, maximum, minimum, and delta frequencies, duration, number of local extrema) were non-parametric (Shapiro-Wilk test; p < 0.05).” (lines 186-189)

9. Figures 2-6: Clearer figures (particularly the labels & values) should be provided. The present Figures are not readable!

• Thank you. We have updated the figures. 

10. Line 149: sqrt transformation, write in full at 1st mention.

• Thank you. We have spelled it out. 

11. Line 152 7 others that follow. What are the units of the frequencies and other parameters?

• Thank you for this question. These numbers are the result of the MANOVA tests, not the values indicating the varied frequencies, therefore it is not appropriate to include units here, but we have referred to them in the Figures. 

12. Lines 295-298: from ‘These findings…..to the end. The statement looks ambiguous and incomplete. Consider rephrasing with the clear meaning of the intended message. In motivating the need for effective management of the ocean due to increasing activities, more details should be provided by citing relevant related literature.

• We have rearranged this section to read: “Increasing sound levels in the ocean may continue to cause changes in dolphin calls leading to information loss (51) and reducing cohesion amongst dolphin groups, including mother-calf pairs, who use signature whistles to facilitate reunions (8). Sound levels in this region will rise due to development of offshore wind energy, coastal development, bridge construction, and increasing vessel traffic. It is particularly important to understand how individuals from particular populations may be exposed to sound from various human activities and hence how they may be impacted by masking of calls and potential disruption to their communication that could affect their fitness. The use of signature whistles to track individuals can also aid in determining how long individuals take to return to an area and resume normal behaviors after exposure to elevated sound levels. Such information will be vital for management of this region to determine the potential effects of, responses to, and recovery from anthropogenic sound pollution (50).” (lines 339-352)

Reviewer #2: I appreciate the dataset that was curated for this manuscript and know all of the hard work that goes into these sorts of large data analyses. Unfortunately, I do not recommend this manuscript for publication at this time. The main reasons are with the methodology: the methods are not the same for all sites, the methods are not described in enough detail for SIGID, Artwarp thresholds, how many of the same whistle type were allowed at each site, the random forrest classifiers used to classify bottlenose from common dolphins whistles, and the low frequency sampling rate assessing things like maximum whistle frequency and broadband ambient noise levels. I think that with some care there can be some interesting findings from the robust dataset if the authors are able to consider these points.

Line Item Edits

Line 35: This is worded very strangely, consider reworking to just say that marine mammals are very vocal.

• Thank you. We have rephrased as suggested. “In the marine environment, one of the most vocal groups of organisms are marine mammals, which produce a variety of calls.” (lines 36-39)

Line 38: Caldwell and Caldwell 1965 founded signature whistles not Sayigh 1990

• We have corrected this, thank you. 

Line 80. Why were all of these methods different? 8:2 and 9:1 duty cycle, visual observations compared to the whistle and moan detector?

Please include your own methods on identifying signature whistles from recordings using SIGID and Artwarp. These methods need to stand alone without needing to refer to other papers for operational definitions and thresholds used. For example what similarity threshold did you use to consider a signature whistle a reoccurrence.

A 48 kHz sampling rate gives a 24kHz Nyquist cut off. Dolphin whistles can range far above this range (up to over 40kHz). This needs to be clearly identified as a limitation of the study and may have resulted in distorted whistle feature data.

• These methods varied because the needs and recording conditions at different sites and time periods varied. The memory/battery capacity of the different recorders that were deployed in the shallower Bay waters compared to the deeper offshore site and the required deployment period between recovery cruises dictated the necessary duty cycle. We have clarified that some sites also had longer deployments, so recording needs varied. 

o “Due to longer deployments at Sites 4 and 5, , the Snap was duty-cycled for two minutes on and eight minutes off at Site 4 and one minute on and nine minutes off at Site 5..” (lines 92-94)

We have also clarified that observers were only present at Site 4, so those criteria were only used for that site. At all sites, methods for narrowing down hours with possible signature whistles were used- whether focusing on the days with visual observations or use of the Whistle and Moan Detector. 

“At Site 4, a subsample of days were analyzed because of the high rate of dolphin presence at this location. At Site 5 (as with Sites 1 and 2), the PAMGUARD Whistle and Moan Detector (19) was utilized to determine hours with possible dolphin presence and these hours were then manually searched for signature whistles with high signal-to-noise ratios.” (lines 94-99) 

• We thank the reviewer- we have included the frequency limitation as a caveat in the Discussion. “Additionally, because sounds with frequencies above 24 kHz could not be detected in our recordings, higher frequency signature whistles would have been missed in our analyses [39].” (lines 318-320)

Line 84 you switch to using numbered citations instead of name and date?

• We have remedied this error. 

Line 94: I am concerned about the species classification portion of this. You say you used ROCCA for species classification but it performed poorly (no data) and then say you used random forest models to confirm species identity but don’t provide any information on those models, their previous success rate or how you confirmed that those were successful here.

When you identified a signature whistle, did you analyze multiple iterations of that whistle or just 1 example? What if one dolphin is hanging around one site for a long time whistling, does their one unqiue signature whistle end up contributing much of the whistle data?

• We understand this concern. We have included text below to elaborate on these results. 

“During the summer at the most inshore site (Site 1), bottlenose dolphins are the most likely species to be detected (25). In recordings from 21 hours on 16 days during the summer at Site 1, the ROCCA algorithm classified 61% of whistles as “Ambiguous” (n = 576), 13% as striped dolphin (Stenella coeruleoalba; n = 121), 11% as bottlenose dolphins (n = 103), 8% as common dolphins (Delphinus delphis, n = 74), 4% as Clymene dolphin (Stenella clymene, n = 39), 2 as Atlantic spotted dolphin (Stenella frontalis, n = 19), and 1% as Pantropical spotted dolphin (Stenella attenuate, n.= 7). Clymene dolphins are typically found in waters deeper than 800 feet, and Pantropical spotted dolphins are not located in the Atlantic Ocean.” (lines 121-129)

• Additionally, we have added text to elaborate our methods so this paper stands alone. “Signature whistles were obtained using the SIGID criteria (1)-the same whistle repeated in a pattern of two or more whistles (7) within 1–10 s of one another and with a minimum length of 0.2 s (20,21). Whistle contours (shape of the whistle) were obtained using Beluga software (https://synergy.st-andrews.ac.uk/soundanalysis) within MATLAB (Math-Works, Natick, Massachusetts, USA). Whistles with low signal-to-noise ratio or abundant non-linear features (22) that obscured the shape of the whistle could not be included in the analysis. Beluga contours were analyzed within the adaptive resonance theory neural network (ART-warp; vigilance threshold of 96%; (23)) to identify when a whistle reoccurred. Identical signature whistles were not considered re-occurrences if they occurred on the same day at the same site. A human analyst verified all whistle re-occurrences.” (lines 99-108)

Line 125- What is a ‘clear’ signature whistle? Sportelli et al 2022 found that over 50% of signature whistles have non-linear features… Does this mean you did not include any whistles that had NLP?

• Whistles with non-linear features were included as long as analysts could determine the shape of the whistle. If too many non-linear features obscured the shape of the whistle, Beluga could not be used to obtain a contour, and thus the analysis could not be completed in ARTwarp. We have added text to indicate this. “Whistles with low signal-to-noise ratio or abundant non-linear features [22] that obscured the shape of the whistle could not be included in the analysis.” (lines 103-104)

Line 129- I am having a hard time with calling the ambient sound levels ‘broad band’ up to 24kHz when dolphin hearing goes into the 100s of kHz. Arguably there could have been any number of anthropogenic sound sources above the recording range.

• In this case, the term broadband was used to indicate that the sound level was calculated over the frequency range of the recorder, not to the hearing range of the species. We have rephrased this to read “as the broadband (up to 24 kHz, given the sampling rate of 48 kHz) sound pressure level” (line 159-161)

Discussion- what was the point of identifying signature whistles if the question was based on whistle features changing based on population, location and or ambient noise? A discussion of why not include all whistles would be helpful.

• Our research questions were: whether signature whistles varied by site indicating differential habitat use by individuals and potentially delineating populations; and whether animals adjusted their identity signature whistles in response to varied sound levels. Since this focuses on these unique, individually identifiable signature whistles, we did not include other types of whistle in our analysis.

If the signature whistle features were significantly different than a random selection of whistles at the same site, wouldn't this be an important result to suggest that the differences between sites may not be due to sites or ambient noise but just based on the fact that the different dolphins have different signature whistles?

• This is an excellent observation. However, because we did not compare a random sample of whistles from each site to signature whistles from those sites, we cannot make the conclusion that signature whistles vary from or are similar to other whistles from those locations or in those sound levels. We specifically compared signature whistles between sites and then examined whether there were any differences in repeated emissions of these signature whistles recorded during reoccurrences at other sites and under different ambient sound conditions.

Line 276: Your recording devices did not go up to the highest recorded whistle frequencies. See Jones et al., 2020 for a review or Kaplan and Reiss 2017 or Hiley et al. 2017 for evidence of whistles into the 40s of kHz.

• Thank you for these citations. We have added a citation to caveat our results. “Additionally, because sound with frequencies above 24 kHz could not be detected in our recordings, higher frequency signature whistles would have been missed in our analyses [35].” (lines 318-320)

---

## [Decision Letter · Decision Letter 1]

17 Nov 2023

PONE-D-23-09444R1Vocal signatures affected by population identity and environmental sound levelsPLOS ONE

Dear Dr. Fandel,

Thank you for submitting your manuscript to PLOS ONE. After careful consideration, we feel that it has merit but does not fully meet PLOS ONE’s publication criteria as it currently stands. Therefore, we invite you to submit a revised version of the manuscript that addresses the points raised during the review process.

We look forward to receiving your revised manuscript.

Kind regards,

Vitor Hugo Rodrigues Paiva, Ph.D.

Academic Editor

PLOS ONE

Reviewers' comments:

Reviewer's Responses to Questions

**Comments to the Author**

1. If the authors have adequately addressed your comments raised in a previous round of review and you feel that this manuscript is now acceptable for publication, you may indicate that here to bypass the “Comments to the Author” section, enter your conflict of interest statement in the “Confidential to Editor” section, and submit your "Accept" recommendation.

Reviewer #1: (No Response)

Reviewer #3: All comments have been addressed

Reviewer #4: (No Response)

2. Is the manuscript technically sound, and do the data support the conclusions?

Reviewer #1: Yes

Reviewer #3: No

Reviewer #4: Yes

3. Has the statistical analysis been performed appropriately and rigorously? 

Reviewer #1: Yes

Reviewer #3: No

Reviewer #4: Yes

4. Have the authors made all data underlying the findings in their manuscript fully available?

Reviewer #1: Yes

Reviewer #3: No

Reviewer #4: Yes

5. Is the manuscript presented in an intelligible fashion and written in standard English?

Reviewer #1: No

Reviewer #3: Yes

Reviewer #4: Yes

6. Review Comments to the Author

Reviewer #1: The authors have made considerable efforts to improve the quality of the previous manuscript. However, there are still some areas that need to be checked/addressed. I have attached 2 documents for the authors' attention.

Reviewer #3: Introduction

Line 32-33: could perhaps use another adjective to ‘challenging’ twice

34-35: These sentences could be edited/combined to help flow

37: detection of individuals and population groups

42: In killer whales (Orcinus orca)

Line 36-45: perhaps consider rearranging so that all the information on calls unique to population groups, and then calls unique to individuals are together

Line 50-52: either move the ‘1)’ to after ‘determine’ or add a verb after the ‘2)’ and ‘3)’

Perhaps a sentence or two could be added about previous work using signature whistles previously, and if they have been used before in this way

Methods

Line 66: What does the Garrod et al reference refer to – make this clear

Line 73, 84: ‘described in 14’?, ‘methodology in 14’

Line 72- 77: these details may be clearer in a table, and is somewhat repetitive to lines 63-66

Line 73: What type of recorders were used at sites 1-3? Even if referencing another study, the basic details should be available to the reader here

Line 79: how was the duty cycle timing selected – was it purely to extend the recording to 2 months

Please be more specific with the timing of the recordings

Line 82: Was any validation of the use of the PAMGUARD W&M detector performed to check accuracy. Were any times when calls were not indicated checked for presence to ensure accuracy and reliability of the detectors? This could be a major limitation in the call identification phase

Line 88-90: were the sites and recording timing chosen because of this or was it coincidental?

Line 91-94: same comment for PAMGUARD applies here for ROCCA – was it tested and verified in any way with this population, given known limitations and poor performance. You mention in lines 94-95 that there is poor performance in this analysis – so the reader needs details about how this was established

Lines 96-97: Much more details on this process is needed. Is there any researcher validation in this process?

Line 101: perhaps good to state what you mean by ‘full state was clear’ means to you

Line 101-104: How were these manual measurements taken?

Line 107: why push into parameteric processes?

Line 112-114: would the random forest not be a better method here throughout and not just to establish input importance? More details of this process are needed, including more details about the input variables and their resolution

Line 126: How did you define ‘changes in ambient sound levels’ – they are constantly changing

Line 128-129: How were ambient sound levels calculated?

Results

Line 147-148: what does ‘become normally distributed’ mean?

Line 154: what does number of extrema mean, and what are you using it to indicate?

Some formatting/text issues on Line 159

Line 170-175: if there are significant differences by site then how would this impact your conclusions based on ambient sound levels etc.

- Are dolphin groups known to be site loyal – what does this result say to your group/individual tracking question

Line 185-187: What do you mean here?

Figure 5: more details in the methods section would help with the interpretation of these results and knowing what was an input variable and what was a shadow variable – and how that was selected and included into the model

How would a variable decrease the accuracy of the model – why would it not just be a none significant/none impactful factor?

Although Figure 6 is informative – I wonder whether the details in the figure and the text (Lines 207-211) would be better displayed in a table or matrix of some kind

Line 218-221: More details are needed here

Discussion

Line 225: This may be true, except you have identified that these unique whistles at different sites – with the calls at the deeper locations also heard at the very coastal locations

Line 226-228: This should be a separate discussion, as this may or may not be site related

Line 229-230: I am not sure how you have demonstrated this with your results as the paper/results are presented currently. Perhaps the site differences and cross over between sites (or not) should be stated more strongly/clearer – and the loyalty of population groups to sites be highlighted from the outset if this is indeed the case (perhaps not given comments in Lines 244-245)

Line 256: why were these sites and photo-ID not used in this study?

Line 170: why was 120 dB used as your threshold for ‘high ambient sound levels’ – perhaps give a reference

Line 273: so whistles may increase with volume when other/more conspecific calls are present? This is a very different conclusion to purely stating ‘increased ambient sound levels’ increased

Line 285: what did you classify as low frequency sounds?

Reviewer #4: Dear author,

I appreciate the work you've done and your addressing the other reviewers comments. That improved the quality of the manuscript. There are some things that are still not clear to the reader:

Here are my comments.

L75 - you should also mention the reference number (18)

L103 - reference 23 should be inside brackets "... 96%; (23)) to ...."

L115- This needs to be further explained. You say your results are poor in what concerns spcs Identification and then just say you relied your certainty of the species using random forest (line 125-127) from a different study on the same populations. Can you explain a bit further on those results? I should not have to go to the reference to look for your study results.

L117- By "In recordings from 21 hours on 16 days" you mean "In a total of 21 hours of recordings from 16 days.."

L122 - It should read "attenuata" instead of "attenuatte"

L133- Should read as "was analyzed" instead of "were analized.

L138 - in the version that I have, Figure 2 is repeated

L149 - This section needs to be further explained and developed. I understand that you compared the SPL levels of those whistles that reocurred at least once. So your N = 100? that's what you used for the GEE?. This needs further explaining in this section.

154-155 - it should read as " Ambient sound levels were calculated in MATLAB (MathWorks,

Natick, Massachusetts, USA) as the relative broadband (up to 24 kHz, given the sampling rate of

48 kHz) root mean square (RMS) sound pressure level (SPL in dB re 1µPa ) during the recording in

which the signature whistle occurred (two or five minutes in duration)"

L306 - maybe a reference here to justify that 120dB is high. Southall et al has good research on this topic.

7. PLOS authors have the option to publish the peer review history of their article (what does this mean?). If published, this will include your full peer review and any attached files.

Reviewer #1: No

Reviewer #3: No

Reviewer #4: No

---

## [Author Response · Author response to Decision Letter 1]

31 Jan 2024

Vocal signatures affected by population identity and environmental sound levels

ID No: PONE-D-23-09444R1

We thank the reviewers and editors for their time and thoughtful reviews of this manuscript. We have included our responses to their comments below. Please let us know if any additional edits are suggested. Thank you again. 

Reviewer's Responses to Questions

Comments to the Author

Reviewer #1: 

The authors have made considerable efforts to improve the quality of the previous manuscript. However, there are still some areas that need to be checked/addressed. I have attached 2 documents for the authors' attention.

Line 34: PAM is especially valuable for monitoring vocalizing species.

 We have rephrased as suggested. 

Line 35: This is not a correct definition of PAM. 

PAM is for data recording/gathering, not for detection.

The readers will benefit/be well carried along with good opening to your paper. 

see https://ieeexplore.ieee.org/abstract/document/9110497 or https://doi.org/10.1016/j.ecoinf.2022.101766 for reference.

 We have corrected our definiteion and re-structured our introduction to flow from group to 

 individual acoustic indentification. We have also added a few lines as a more compelling 

 introduction.

Line 47: this sentence is unclear/incomplete. Kindly look into it.

 We have rephrased to read “Distinct populations of bottlenose dolphins vary the 

characterstics of their whistles (14,15), but variations in signature whistles have not yet 

been investigated.” (line 54-57) 

Line 54: Make your manuscript reading flow.....This study aims to 1) to determine ..aims to 2) whether? it should be 2)know/determine whether there are..... The same to the 3rd aim.

We have rephrased to read “This study aims to determine 1)…” 

Line 66: Does this mean anyone can just put autonomous recorders in the study area?

 For the majority of the area, that is true. We have clarified “No permits were required for 

this work as there was no handling or interaction with the study species, but permission 

was obtained from the leaseholders of the wind energy area for the deployment of 

bottom-anchored autonomous recorders.” (line 77-80)

Line 105: check you reference format and other formating style.

 Thank you. Corrected. 

Line 114: is this a reference?

 This is a reference to personal communications. The style has been corrected. 

 

Reviewer #3: 

Introduction

Line 32-33: could perhaps use another adjective to ‘challenging’ twice

Apologies, I do not see the duplicated word here.

34-35: These sentences could be edited/combined to help flow

We have combined them as suggested. 

37: detection of individuals and population groups

Changed as requested

42: In killer whales (Orcinus orca)

Changed as requested

Line 36-45: perhaps consider rearranging so that all the information on calls unique to population groups, and then calls unique to individuals are together

Revised as requested. Thank you. 

Line 50-52: either move the ‘1)’ to after ‘determine’ or add a verb after the ‘2)’ and ‘3)’

Moved determine to before the 1) as suggested. 

Perhaps a sentence or two could be added about previous work using signature whistles previously, and if they have been used before in this way

We have added a sentence outlining previous studies in this region. “Previous studies in 

this region investigated the presence and temporal patterns of bottlenose dolphins (17–20) and documented their signature whistles (21). This study aims to build upon that work and determine…” (line 61-63)

Methods

Line 66: What does the Garrod et al reference refer to – make this clear

We have removed this reference.

Line 73, 84: ‘described in 14’?, ‘methodology in 14’

We have revised line 73, and removed the reference from line 84. 

Line 72- 77: these details may be clearer in a table, and is somewhat repetitive to lines 63-66

 We have added a table as suggested (Table 2).

Line 73: What type of recorders were used at sites 1-3? Even if referencing another study, the basic details should be available to the reader here

 Details on recorders have been included in Table 2. 

Line 79: how was the duty cycle timing selected – was it purely to extend the recording to 2 months

Please be more specific with the timing of the recordings

 We have specified that duty cycling was selected simply to prolong battery life. 

Deployments were 3-6 months. 

Line 82: Was any validation of the use of the PAMGUARD W&M detector performed to check accuracy. Were any times when calls were not indicated checked for presence to ensure accuracy and reliability of the detectors? This could be a major limitation in the call identification phase

 After the PAMGUARD W&M detector indicated dolphin presence, he process of 

selecting the signature whistles was entirely manula, so we are positive that these are true 

detections. We have clarified this detail.

“Signature whistles were manually identified using the SIGID criteria” (line 115-117)

Line 88-90: were the sites and recording timing chosen because of this or was it coincidental?

 Yes, we had prior knowledge that dolphins were present in this region at these times. 

Line 91-94: same comment for PAMGUARD applies here for ROCCA – was it tested and verified in any way with this population, given known limitations and poor performance. You mention in lines 94-95 that there is poor performance in this analysis – so the reader needs details about how this was established

 ROCCA was only used to determine whether it was a viable option for determining 

which species were present in the recordings. However, it was found to be extremely 

unreliable and not used in the final analyses. Including this analysis was recommended by 

another reviewer. 

Lines 96-97: Much more details on this process is needed. Is there any researcher validation in this process?

 As mentioned above, including this analysis was only included to document that using 

ROCCA was not a viable option in determining which species was present. 

Line 101: perhaps good to state what you mean by ‘full state was clear’ means to you

 We have removed this to improve clarity. 

Line 101-104: How were these manual measurements taken?

 We have clarified that these measurements were taken in Raven. Markers were manually 

placed on the whistle. 

Line 107: why push into parameteric processes?

 We wanted to use a MANOVA to test whether the characteristics of whistles varied by 

site, which are parametric analyses. 

Line 112-114: would the random forest not be a better method here throughout and not just to establish input importance? More details of this process are needed, including more details about the input variables and their resolution

 ROCCA is an algorithm that can be used in PAMGUARD. We have added additional 

details on our analyses and rearranged for clarity. “To confirm that the species detected 

were bottlenose, not common dolphin (Delphinus delphis) ), we tested the utility of 

the Real-time Odontocete Call Classification Algorithm (ROCCA) (28) in 

PAMGUARD.” (line 131-134)

Line 126: How did you define ‘changes in ambient sound levels’ – they are constantly changing

 The sound level during the recording (2-5 minutes) was included in the GEE model and 

modeled against the signature whistle characteristics. We have rephrased: “Generalized 

estimating equations-generalized linear models (GEEs) in R (geepack package in R; (33) 

were used to determine whether and which changes in signature whistles characteristics 

were correlated with changes in ambient sound levels.” (line 193-196)

Line 128-129: How were ambient sound levels calculated?

 “Ambient sound levels were calculated in MATLAB (MathWorks, Natick, 

Massachusetts, USA) as the relative broadband (up to 24 kHz, given the sampling rate of 

48 kHz) sound pressure level (SPL; dB re 1µPa root-mean-square (rms)) during the 

recording in which the signature whistle occurred (two or five minutes in duration).” (line 

188-191)

Results

Line 147-148: what does ‘become normally distributed’ mean?

 We have rephrased to read “were normally distributed”. (line 218) We meant to indicate 

whether the characteristics were normally distributed after transformation.

Line 154: what does number of extrema mean, and what are you using it to indicate?

 The number of extrema is a count of the local minima and maxima and can be considered 

a proxy for complexity. We have clarified “A MANOVA indicated that start (F4,352 = 

2.71, p = 0.03), end (F4,352 = 4.52, p = 0.01), and minimum frequencies (F4,352 = 10.3, p < 

0.01) as well as duration (F4,352 = 9.36, p < 0.01) and number of extrema (F4,352 = 7.28, p 

< 0.01), or complexity, of signature whistles varied significantly by site.” (line 224-227)

Some formatting/text issues on Line 159

 I believe this is a result of Word’s formatting. It should be fixed when formatted by the 

Journal. 

Line 170-175: if there are significant differences by site then how would this impact your conclusions based on ambient sound levels etc.

- Are dolphin groups known to be site loyal – what does this result say to your group/individual tracking question

 Dolphins are generally not loyal to specific sites in this region, though offshore 

bottlenose dolphins do not generally come to more inshore sites. 

Line 185-187: What do you mean here?

 We have struck this portion, as we believe that we indicated our methods in the sentences 

before in which we state the transformations of the variables. 

Figure 5: more details in the methods section would help with the interpretation of these results and knowing what was an input variable and what was a shadow variable – and how that was selected and included into the model

 I have added details in the methods to read: “The Boruta package in R (32) randomizes 

the data for each factor (called shadow data) then utilizes random forest models with the 

data and shadow data to assess factor importance in determining an identity (e.g. site or 

region).” (line 178-181)

How would a variable decrease the accuracy of the model – why would it not just be a none significant/none impactful factor?

 We have rephrased to “When the start frequency and number of extrema were included in 

the model, the model was less accurate when determining the region to which a signature 

whistle belonged (Fig 5b).” (line 257-260)

Although Figure 6 is informative – I wonder whether the details in the figure and the text (Lines 207-211) would be better displayed in a table or matrix of some kind

 We have used this plot to illustrate the variation in sound levels at each site and whether 

the sounds levels varied significantly between each. We would prefer to keep this plot. 

Line 218-221: More details are needed here

We have added “Reoccurences of whistles indicate that dolphins were detected at 

different sites on different days.” (line 279-280) to clarify the significance of 

reoccurences. 

Discussion

Line 225: This may be true, except you have identified that these unique whistles at different sites – with the calls at the deeper locations also heard at the very coastal locations

 We have added additional details to the first sentence. “We determined that bottlenose 

dolphins reoccurred through time and space and that the characteristics of bottlenose 

dolphin signature whistles significantly varied by site and region.” (310-311)

Line 226-228: This should be a separate discussion, as this may or may not be site related

 We have expanded upon this below, but wanted to provide an overview of the discussion. 

Line 229-230: I am not sure how you have demonstrated this with your results as the paper/results are presented currently. Perhaps the site differences and cross over between sites (or not) should be stated more strongly/clearer – and the loyalty of population groups to sites be highlighted from the outset if this is indeed the case (perhaps not given comments in Lines 244-245)

 We have expanded upon the reocurrences of the whistles between sites. 

“In total, 201 unique signature whistles reoccurred a total of 252 times at the five 

different sites (Fig. 7). Reoccurences of whistles indicate that dolphins were detected at 

different sites on different days. Signature whistles most often reoccurred at the inshore sites, Sites 1 and 2 (n = 49; 15% of all unique signature whistles from Site 2) and between the most inshore site, Site 1, and the lower Bay site, Site 4 (n = 30, 9% of unique signature whistles from Site 4; Fig. 3.9).

Of the 19 unique signature whistles detected at the upper Bay site (Site 3), 26% (n = 5) were also detected at the lower Bay site (Site 4;Fig. 7). At the lower Bay site, 24 of the total 333 unique signature whistles identified reoccurred 27 times within the site on different days indicating that dolphins may be spending several days within the Bay and were detected as they enter and exit the Bay. 

Of the 37 unique signature whistles detected at Site 5, only three signature whistles reoccurred within the site. Signature whistles from the offshore Site 5 were also detected three times at Site 2 (two of which were the same whistle; 5% of all unique signature whistles from Site 5) and one was detected at Site 1 (3%; Fig. 7). “ (line 278-292)

Line 256: why were these sites and photo-ID not used in this study?

 Photo-ID was not available for this study, and acoustic recordings were not available 

when the study with photo-ID were conducted. 

Line 170: why was 120 dB used as your threshold for ‘high ambient sound levels’ – perhaps give a reference

 We have added a reference to the National Research Council’s 2005 paper on 

biologically significant effects of noise.

Line 273: so whistles may increase with volume when other/more conspecific calls are present? This is a very different conclusion to purely stating ‘increased ambient sound levels’ increased

 We did not assess the volume of the calls, only the ambient sound level in the two or five 

minute recording in which the signature whistle occurred. Thus, ambient sound levels in 

these recordings could be due to other whistles during the recording. We have added text 

to read “Because our calculation of ambient sound levels did not exclude other whistles, 

higher sound levels could also be indicative of more whistles within the 2 or 5-minute 

recordings.” (line 358-360)

Line 285: what did you classify as low frequency sounds?

 These studies reference fish calls and sounds below the frequencies of bottlenose dolphin 

whistles, so we have specified in the text. “Whistles with higher frequencies (27,41) and 

fewer extrema (27,40) are common where low frequency (generally below 3 kHz) 

ambient sound levels are higher.” (line 370-372)

Reviewer #4: 

L75 - you should also mention the reference number (18)

 Added. Thanks!

L103 - reference 23 should be inside brackets "... 96%; (23)) to ...."

 Should be corrected now. 

L115- This needs to be further explained. You say your results are poor in what concerns spcs Identification and then just say you relied your certainty of the species using random forest (line 125-127) from a different study on the same populations. Can you explain a bit further on those results? I should not have to go to the reference to look for your study results.

 We have added clarification that we used the spatial and temporal patterns of species’ 

presence determined in previous studies. “Instead of relying on these automated detection 

methods, we used the spatial and temporal patterns determined in previous studies 

(20, 29) to determine which species were present and avoided sites and seasons in which 

there was likely to be overlap in presence.” (line 154-157)

L117- By "In recordings from 21 hours on 16 days" you mean "In a total of 21 hours of recordings from 16 days.."

 Corrected.

L122 - It should read "attenuata" instead of "attenuatte"

 Corrected. 

L133- Should read as "was analyzed" instead of "were analized.

 Changed as suggested. Thank you. 

L138 - in the version that I have, Figure 2 is repeated

 Apologies. There is only one Figure 2 included. 

L149 - This section needs to be further explained and developed. I understand that you compared the SPL levels of those whistles that reocurred at least once. So your N = 100? that's what you used for the GEE?. This needs further explaining in this section.

 We have clarified our methods here and added “The characteristics of 100 randomly 

selected signature whistles that re-occurred were modeled with the GEEs against the

sound levels in which they occurred.” (line 196-197)

154-155 - it should read as " Ambient sound levels were calculated in MATLAB (MathWorks,

Natick, Massachusetts, USA) as the relative broadband (up to 24 kHz, given the sampling rate of

48 kHz) root mean square (RMS) sound pressure level (SPL in dB re 1µPa ) during the recording in which the signature whistle occurred (two or five minutes in duration)"

 This change has been made as suggested (line 190).

L306 - maybe a reference here to justify that 120dB is high. Southall et al has good research on this topic.

 We have added a reference to the National Research Council’s 2005 paper on biologically significant effects of noise.

---

## [Editor Report · Decision Letter 2]

8 Feb 2024

Vocal signatures affected by population identity and environmental sound levels

PONE-D-23-09444R2

Dear Dr. Fandel,

We’re pleased to inform you that your manuscript has been judged scientifically suitable for publication and will be formally accepted for publication once it meets all outstanding technical requirements.

Kind regards,

Vitor Hugo Rodrigues Paiva, Ph.D.

Academic Editor

PLOS ONE
---

## [Editor Report · Acceptance letter]

24 Mar 2024

PONE-D-23-09444R2 

PLOS ONE

Dear Dr. Fandel, 

I'm pleased to inform you that your manuscript has been deemed suitable for publication in PLOS ONE. Congratulations! Your manuscript is now being handed over to our production team.

Kind regards, 

on behalf of

Dr. Vitor Hugo Rodrigues Paiva 

Academic Editor

PLOS ONE